# Metabolic and Transcriptional Analysis Reveals Flavonoid Involvement in the Drought Stress Response of Mulberry Leaves

**DOI:** 10.3390/ijms25137417

**Published:** 2024-07-06

**Authors:** Guo Chen, Dong Li, Pei Yao, Fengyao Chen, Jianglian Yuan, Bi Ma, Zhen Yang, Biyue Ding, Ningjia He

**Affiliations:** 1State Key Laboratory of Resource Insects, Institute of Sericulture and Systems Biology, Southwest University, Chongqing 400715, China; q542523966@163.com (G.C.); lidong203@swu.edu.cn (D.L.); wanmbbmmdd@email.swu.edu.cn (P.Y.); cfy0127@email.swu.edu.cn (F.C.); yuanjiangl@swu.edu.cn (J.Y.); mbzls@swu.edu.cn (B.M.); yangzhen1246305364@163.com (Z.Y.); 2Key Laboratory of Agricultural Biosafety and Green Production of Upper Yangtze River (Ministry of Education), Academy of Agricultural Sciences, Southwest University, Chongqing 400715, China; dingbiyue302@swu.edu.cn

**Keywords:** drought stress, mulberry, flavonoid biosynthesis, diversified utilization

## Abstract

Abiotic stress, especially drought stress, poses a significant threat to terrestrial plant growth, development, and productivity. Although mulberry has great genetic diversity and extensive stress-tolerant traits in agroforestry systems, only a few reports offer preliminary insight into the biochemical responses of mulberry leaves under drought conditions. In this study, we performed a comparative metabolomic and transcriptomic analysis on the “drooping mulberry” (*Morus alba* var. *pendula* Dippel) under PEG-6000-simulated drought stress. Our research revealed that drought stress significantly enhanced flavonoid accumulation and upregulated the expression of phenylpropanoid biosynthetic genes. Furthermore, the activities of superoxide dismutase (SOD), catalase (CAT) and malondialdehyde (MDA) content were elevated. In vitro enzyme assays and fermentation tests indicated the involvement of flavonol synthase/flavanone 3-hydroxylase (XM_010098126.2) and anthocyanidin 3-*O*-glucosyltransferase 5 (XM_010101521.2) in the biosynthesis of flavonol aglycones and glycosides, respectively. The recombinant MaF3GT5 protein was found to recognize kaempferol, quercetin, and UDP-glucose as substrates but not 3-/7-*O*-glucosylated flavonols and UDP-rhamnose. MaF3GT5 is capable of forming 3-*O*- and 7-*O*-monoglucoside, but not di-*O*-glucosides, from kaempferol. This implies its role as a flavonol 3, 7-*O*-glucosyltransferase. The findings from this study provided insights into the biosynthesis of flavonoids and could have substantial implications for the future diversified utilization of mulberry.

## 1. Introduction

Terrestrial plant growth is strongly influenced by various abiotic stresses, including drought, temperature, alkalinity, radiation, and salinity. Among these stressors, drought is recognized as the most severe limiting factor affecting plant survival and yield [1]. Meanwhile, drought represents a persistent global ecological challenge exacerbated by increasing temperatures [2]. Drought conditions typically lead to the accumulation of excess reactive oxygen species (ROS), reduced photosynthesis, and disruptions in plant metabolism [3,4]. In order to adapt to water scarcity, plants have developed various biological strategies, such as biochemical responses [5], antioxidant defense mechanisms [6], and the regulation of stomatal movement [7]. Numerous studies have revealed the differences in drought resistance among species and varieties, directly influencing factors such as total biomass, leaf area, and the root-to-shoot ratio [8,9,10]. Therefore, a comprehensive understanding of the mechanisms governing plant responses to drought stress is crucial for the effective screening and breeding of drought-resistant varieties.

The mulberry (*Morus* spp.) is an important tree crop in agroforestry, yielding fodder, fruit, fuel, and wood. It has been widely cultivated for thousands of years in China [11]. Mulberry leaves are used not only as forage for silkworms in sericulture but also as herbal medicine and functional nutraceutical food for humans [12,13]. For a long time, growers have traditionally focused on the economic and pharmacological benefits due to the richness of mulberry leaves in various bioactive compounds like flavonoids, alkaloids, polyphenols, polysaccharides, and vitamins [14,15]. However, the growth and productivity of mulberry leaves are remarkably influenced by drought stress. Despite the genetic diversity of mulberry and its stress-tolerant traits in agroforestry systems [16], there is limited information available on the biochemical responses of mulberry leaves to drought conditions [17]. The potential ecological benefits of mulberry leaves remain largely unexplored [18], promising diversified utilization of this valuable “food and medicine homologous agricultural resource”.

Flavonoids are considered among the most critical constituents in mulberry leaves due to their diverse biochemical and pharmacological properties [19,20]. The biosynthesis of flavonoids is derived from phenylpropanoid metabolism and is regulated by a series of key genes [21]. In a study involving prolonged drought treatment in *Achillea*, it was observed that the expression of phenylalanine ammonia lyase (*PAL*) and flavone 3-hydroxylase (*F3H*) was significantly elevated at the onset of stress, later decreasing in the middle stage, whereas the expression of chalcone synthase (*CHS*), chalcone isomerase (*CHI*), and flavonoid 3′-hydroxylase (*F3*′*H*) increased during the middle stage. Toward the end of the stress period, the expression of all genes, except *PAL*, further increased, with the expression of these genes notably higher under stress conditions compared to non-stress conditions [22]. Recent studies have confirmed that flavonoids can enhance drought tolerance in maize (*Zea mays*) [23], sea buckthorn (*Hippophae rhamnoides*) [9], and wheat (*Triticum aestivum*) [24]. Mulberry leaves are rich in glycosylated flavonols such as astragalin, isoquercitrin, and rutin. Previous research has indicated a significant increase in rutin levels in drought-stressed Fava d’anta (*Dimorphandra mollis*) leaves, suggesting a potential role for rutin in protecting tissues against oxidative damage during drought periods [25].

The antioxidant and free-radical-scavenging ability of flavonoids is attributed to their hydroxyl groups, double bounds, and their predisposition to glycosylation, acylation, and methylation [26,27,28]. Yang et al. [29] reported that drought stress triggers the biosynthesis of flavonoids in *Bupleurum chinense* leaves. Jan et al. [30] also observed that over-accumulation of non-enzymatic antioxidant flavonoids enhanced drought and UV radiation stress tolerance in rice. Recently, a basic leucine zipper (bZIP) transcription factor, *FlbZIP12* from *Fagopyrum leptopodum*, was shown to enhance drought tolerance by modulating flavonoid biosynthesis [31]. Despite previous reports identifying 44 flavonoids in mulberry leaves and offering insights into rutin biosynthesis [13,32], little is known about their evaluation regarding drought stress tolerance. Given that comparative transcriptome analysis has revealed numerous key genes and pathways involved in stress response [33,34,35], a comprehensive, widely targeted metabolic profiling of mulberry leaves could provide further insights into the biochemical changes under drought conditions [36,37].

The primary objective of this study was to elucidate drought-induced key genes or pathways in mulberry leaves and to explore the mechanisms governing flavonoid accumulation under drought stress conditions. To achieve this, we performed a comprehensive analysis of the metabolome and transcriptome, comparing drought-stressed “drooping mulberry” (*Morus alba* var. *pendula* Dippel) leaves with untreated controls as part of our investigation into mulberry drought biology. Our findings indicate significant alterations in phenylpropanoid metabolism and flavonoid content in response to drought stress. Furthermore, we conducted in vitro enzyme assays and fermentation tests to characterize flavonol synthase1 (MaFLS1) and flavonol 3, 7-*O*-glucosyltransferase (MaF3GT5). These enzymes play a pivotal role in the flavonol biosynthetic pathway responsible for synthesizing flavonol aglycones and glycosides [38]. Our research will contribute to a better understanding of the biochemical mechanisms underlying mulberry drought tolerance, potentially facilitating the enhancement of flavonoid content and the medicinal utilization of mulberry resources.

## 2. Results

### 2.1. Measurement of Physiological Indices Related to Drought Damage

Under drought conditions, the accumulated free radicals and active oxygen are closely related to the activity of protective enzymes in plants, consequently leading to lipid peroxidation. In the present study, three key physiological indicators associated with drought stress were assessed: MDA content, SOD activity, and CAT activity. All measured parameters were compared to the control group (0 h). The results revealed that CAT and SOD activities increased under drought stress compared to the control, showing a continuous increase from 0 h to 72 h, with the peak levels observed at 72 h. Specifically, CAT activity rose from 1816.1 U/g at the control (0 h) to 8089.9 U/g at 72 h, while SOD activity increased from 34.2 U/g at the control to 181.2 U/g at 72 h. Meanwhile, MDA concentration also exhibited an upward trend during drought stress. Unlike CAT and SOD, MDA levels increased steadily from the beginning of the experiment (28.5 nmol/g) up to 48 h (51.3 nmol/g), followed by a slight decrease at 72 h (43.91 nmol/g) (Figure 1).

### 2.2. Changes in Flavonoid Level under Drought Stress

The metabolic profiling analysis of mulberry leaves was conducted with three biological replicates under control (0 h) and following 24, 48, and 72 h of drought stress. Utilizing the mulberry metabolome database (MMHub, https://biodb.swu.edu.cn/mmdb/ (accessed on 11 March 2020)) established in a previous study [32], a total of 44 flavonoids present in mulberry and two authentic standards (kaempferol and quercetin) were assessed (Appendix A). In this investigation, a total of 32 flavonoids were identified in *M. alba* var. *pendula* leaves. *O*-glycosylated flavonols and malonylated flavonol glycosides were the predominant compounds. Moreover, some flavones, such as apigenin 7-*O*-glucoside and luteolin 7-*O*-glucoside, were also detected (Appendix A).

The drought stress elicited varied responses in the individual compounds analyzed via UHPLC. Initially, at 0 h (control), the malonylated, mono- and di-*O*-glycosylated flavonols, such as kaempferol 3-*O*-malonylglucoside, quercetin 3-*O*-glucoside, and rutin, exhibited the highest accumulation, while flavones and multiple *O*-glycosylated flavonols were at notably low levels (Figure 2A). Subsequently, at 24 h, all flavonoids showed increased levels compared to the control, with fold changes (log_2_FC) ranging from 0.31 to 9.26. Notably, quercetin *O*-rhamnosyl-*O*-hexosyl-*O*-hexoside-II experienced the most significant increase, while naringenin exhibited the least change. Moving to 48 h, approximately half of the compounds decreased in concentration compared to the levels observed at 24 h. Particularly, quercetin *O*-hexosyl-*O*-hexosyl-*O*-hexoside displayed the most significant decrease with a log_2_FC of −3.87. Moreover, certain flavonoids like apigenin maintained a continuous increase (log_2_FC = 3.70), while others showed no significant change. Almost all flavonoids reached their peak concentration at 72 h. Quercetin *O*-rhamnosyl-*O*-hexosyl-*O*-hexoside-I and naringenin (0.12 and 0.17, respectively) showed no change compared to 48 h, whereas kaempferol *O*-rhamnosyl-*O*-malonylhexoside exhibited the most significant increase with a log_2_FC of 4.15 (Figure 2B).

### 2.3. Identification of Flavonoid-Related Candidate Genes through Transcriptome Analysis

In order to identify the key genes related to the biosynthesis of flavonoids or involved in drought stress response in mulberry leaves, we performed an RNA sequencing (RNA-seq) analysis with three replications of two experimental groups: the control (C) and drought-treated at 24 h (D). In the present study, a total of 56,354,744 bases were assembled into 45,642 unigenes, displaying a mean length of 1234 nucleotides (nt), an N50 length of 2292 nt, an N50 count of 7996, and a GC content of 40.44% (Appendix A). The length distribution of all unigenes ranged from 200 to over 3000 nt. Furthermore, 62.2% (28,401) of the unigenes were annotated through BLASTx searches (E-value threshold, 1 × 10^−5^) against the NR, Swiss-Prot, COG/KOG, and KEGG databases, resulting in 27,603, 18,371, 15,845, and 25,378 unigenes, respectively (Appendix A).

We further performed the principal component analyses (PCA) on both the metabolome and transcriptome data. For the metabolome data, PC1 (47.7%) and PC2 (21.1%) collectively explained 68.8% of the variation in the metabolic profile (Appendix A). Similarly, the PCA of transcriptome analysis indicated two distinct clusters along PC1, confirming the adequacy of our model (Appendix A). Moreover, we identified 4530 DEGs, with 2495 genes showing upregulation and the remainder displaying downregulation (Appendix A). Our KEGG pathway analysis revealed that the DEGs primarily participated in metabolism, genetic information processing, environmental information processing, cellular processes, and organismal systems, accounting for 60.8%, 30.2%, 4.6%, 1.6%, and 2.7% of the total, respectively (Appendix A).

The drought-treated group exhibited a significantly higher relative abundance of flavonoids compared to the control group. Subsequently, an in silico filtration process [13,39] was employed to identify candidate genes related to flavonoids within the transcriptome libraries. In the present study, we identified 18 genes significantly changed in the flavonoid pathway (Appendix A). Specifically, the expression levels of three *PAL*, two *4CL*, two *CHS*, one *F3H*, two *F3′H*, one *FLS*, and four *UGT* genes displayed a 1.61- to 9.88-fold increase, while the expression of one *CHS* and two *FLS* genes decreased (Appendix A). Moreover, *CHI* and *C4H* did not exhibit significant changes in the transcriptome data. Given that flavonols are the most abundant flavonoids in mulberry leaves and certain key genes related to flavonol metabolism (e.g., *MaF3GT* (KP455729) [40], *MaF3G6″RT* (KT324624) [13], and *MaF3H* (EXC35356.1) [18]) have been characterized in mulberry, we selected four candidate genes for further characterization—*MaFLS1*, *MaUGT76A2*, *MaUGT90A1*, and *MaF3GT5*—due to their high amino acid sequence identity (>64%) with other sources (Appendix A). Furthermore, phylogenetic analysis suggested that *MaUGT76A2* and *MaUGT90A1* probably function as flavonoid rhamnosyltransferases, contributing to the biosynthesis of common *O*-rhamnosylated flavonols in mulberry leaves (e.g., kaempferol 3-*O*-rutinoside (K3R) and quercetin 3-*O*-rutinoside (Rutin)). Moreover, the distinct positioning of *MaF3GT5* in the phylogenetic tree indicated functional differentiation of this candidate gene (Appendix A). Based on these results, we present a proposed flavonoid pathway and the key genes involved in the drought response of mulberry leaves in Figure 3.

### 2.4. Expression Patterns of Phenylpropanoid Biosynthetic Genes under Drought Stress

We conducted a detailed analysis of the expression profiles of phenylpropanoid biosynthetic genes under prolonged drought conditions, focusing on 14 genes involved in the flavonoid pathway and three putative *MaUGTs* identified from the transcriptome data. Additionally, we examined a key enzyme in mulberry flavonol metabolism, flavonol 3-*O*-glucoside: 6″-*O*-rhamnosyltransferase (MaF3G6″RT), as previously described [13]. The qRT-PCR results revealed significant fluctuations in gene expression patterns in response to drought stress (Figure 4). Specifically, the expression levels of *MaPAL1*, *MaPAL2*, and *MaPAL3* were markedly upregulated after 24 h of drought treatment, followed by a significant decrease at 48 h and then stabilization until 72 h. Consistent trends were observed for *Ma4CL1*, *MaCHS1*, and *MaF3GT* over the same time course. Notably, *MaC4H*, *MaCHI*, and *MaF3G6*″*RhaT* showed no significant changes throughout the experiment. On the other hand, *Ma4CL2* and *MaFLS1* exhibited a significant increase at 24 h, followed by a gradual decline until 72 h. *MaCHS2* and *MaF3H* showed a similar pattern with an initial increase at 24 h followed by a plateau until 72 h. In contrast, *MaF3′H1* displayed a consistent increase in expression levels during the entire experiment. Among the candidate *MaUGTs*, *MaUGT90A1* initially increased, then decreased, and subsequently rose again, while *MaUGT76A2* initially showed a slight increase, followed by a significant rise before returning to levels comparable to 24 h (Figure 4). Noteworthy, all phenylpropanoid biosynthetic genes, except *MaC4H2*, *MaCHI2*, *MaFLS2*, *MaF3G*″*6RT*, and *MaUGT76A2*, displayed significant changes in expression at 24 h post-drought treatment, supporting the robustness of the RNA-seq data (Appendix A).

### 2.5. Characterization of Recombinant MaFLS1 and MaF3GT5 Enzymes

The recombinant putative *Morus alba* L. flavonol synthase1 (MaFLS1) protein was successfully expressed in *E. coil* as a His tag-fused protein and then purified for enzymatic assays. MaFLS1 exhibited significant amino acid sequence homology with FLS proteins from various species, such as *Arabidopsis thaliana*, *Malus pumila*, and *Rosa multiflora* (Appendix A). It belongs to the typical 2-oxoglutarate-dependent dioxygenases (2-ODDs) family, which emphasizes its absolute dependence on iron (II), 2-oxoglutarate, and ascorbate for in vitro enzymic activity (Appendix A). The recombinant MaFLS1 protein catalyzed the conversion of dihydroquercetin (7.35 min) and dihydrokaempferol (8.31 min) into distinct products (N1 and N2, respectively) (Figure 5). In the UHPLC-MS/MS analysis, N1 and N2 displayed distinct peaks and molecular ions at a mass-to-charge ratio of 303.0501 [M + H]^+^ and 287.0553 [M + H]^+^, respectively, consistent with the retention time and mass calculation (│*m/z* error│ ≤ 10 ppm) of quercetin (C_15_H_10_O_7_, 303.0499 [M + H]^+^, 9.45 min) and kaempferol (C_15_H_10_O_6_, 287.0550 [M + H]^+^, 10.62 min) (Figure 5A). Furthermore, the mass spectra of N1 and N2 exhibited identical fragmentation patterns to those of the authentic standards of kaempferol (153.0185, 121.0288, and 68.9981) and quercetin (153.0186, 137.0236, 257.0450, and 229.0496) (Figure 5B). In addition, the fermentation liquor of MaFLS1 changed in color from milky white to yellow after 48 h of fermentation at 25 °C (Figure 5A). It is worth noting that the detected fermented products were consistent with the result of the enzymatic assay described above. These observations indicated that *MaFLS1* encodes a flavonol synthase that recognizes dihydroquercetin and dihydrokaempferol as potential substrates in mulberry.

Three recombinant *Morus alba* L. flavonoid-related UGTs were also obtained using the same method as MaFLS1. Two sugar donors (UDP-glucose and UDP-rhamnose) and six typical mulberry leaf flavonol aglycones or glycosides were tested as potential substrates, including kaempferol, quercetin, kaempferol 3-*O*-glucoside, kaempferol 7-*O*-glucoside, quercetin 3-*O*-glucoside, and quercetin-7-*O*-glucoside. Among these, only the recombinant MaF3GT5 protein catalyzed the conversion of kaempferol and quercetin into three products (F1, F2, and F3) (Figure 6). F2 and F3 had distinct peaks and molecular ions with a mass-to-charge ratio of 449.1070 [M + H]^+^ and 449.1068 [M + H]^+^, respectively, consistent with the retention time and mass calculation (│*m/z* error│ ≤ 10 ppm) of kaempferol 3-*O*-glucoside (C_21_H_20_O_11_, 7.66 min) and kaempferol 7-*O*-glucoside (C_21_H_20_O_11_, 7.89 min), as shown in Figure 6. The MS^2^ spectra of F2 and F3 also displayed identical fragmentation profiles to those of the authentic standards of kaempferol 3-*O*-glucoside and kaempferol 7-*O*-glucoside, in which Y_0_^+^ ions were observed at *m/z* 287.0542 and 287.0544 due to the loss of the glucose moiety from the *O*-glycosyl group of the phenolic hydroxyl (−162) (Figure 6). Remarkably, although F1 had a single peak and molecular ions at a mass-to-charge ratio of 465.1027 [M + H]^+^, consistent with the retention time, mass calculation, and MS^2^ spectra of quercetin 7-*O*-glucoside (C_21_H_20_O_12_, 7.09 min), 3-*O*- and 7-*O*-glycosylated quercetin presented the same elution profile (RT^Δ^ = 0.01 min) under our conditions (Figure 6). The recombinant MaF3GT5 protein was found to recognize flavonol aglycones and UDP-glucose but not flavonol glycosides or UDP-rhamnose. These observations indicated that MaF3GT5 functions as a flavonol 3, 7-*O*-glucosyltransferase.

## 3. Discussion

Abiotic stress, particularly drought, can trigger a burst of ROS and affect the metabolic processes of plants, leading to an increased accumulation of H_2_O_2_ and other hydroxyl radicals, which seriously threaten plant growth, development, and production. SOD and CAT are key enzymes within the protective enzymes system and play a pivotal role in scavenging ROS [41]. Moreover, it is likely that SOD and CAT also synergistically interact with protective secondary metabolites to uphold an optimal redox balance and individual fitness in plants [5]. Our study observed a continuous increase in the activities of SOD and CAT during PEG-6000 treatment, thus indicating the reliability of simulating drought stress. Drought stress invariably leads to lipid peroxidation, resulting in the accumulation of MDA. The MDA content serves as an indicator of stress severity. Although the MDA accumulation presented a certain degree of decrease at 72 h, the content of MDA was significantly higher than 0 h (no stress condition) in our research. A possible explanation for the decrease in MDA might be due to the increase in the content of secondary metabolites [42] and the efficient work of ROS detoxifying enzymes [43], especially CAT, whose activity constantly increases during the tested time period.

The 32 flavonoids detected in this study (Appendix A) can be categorized as one flavanone, five flavones, and 26 flavonols, which are generally consistent with the known composition of flavonoids in *M. alba* var. *pendula* leaf reported before [13]. In our study, compared to the untreated counterparts (0 h), PEG-6000 treatment significantly increased the accumulation of flavonoids, potentially reducing ROS levels and enhancing mulberry leaf adaptability to drought stress. Our transcriptome analysis further revealed significant differences in the expression of key genes involved in flavonoid biosynthesis pre- and post-drought treatment, suggesting that flavonoids play a role in the drought tolerance of mulberry. Flavonoids have attracted more attention due to their diverse pharmacological effects. Controlled drought stress appears capable of enhancing the abundance of flavonoids, which could hold considerable medicinal value and broaden the utilization of mulberry leaves.

Through metabolome and transcriptome analysis performed in both drought-stressed and unstressed mulberry, a more comprehensive understanding of the response of the phenylpropanoids pathway to drought was obtained. The drought stress activated the biosynthesis of phenylpropanoids in mulberry leaves. However, as naringenin is considered an intermediate and rapidly converted [44] as the common precursor of a large number of other downstream secondary metabolites like flavones and flavonols, it is logical that its content remained at a relatively low and almost unchanged level (│log_2_FC│ < 0.35) throughout. Similarly, the detection of flavonols presented a comparable situation. While mulberry leaves have abundant *O*-glycosylated flavonols and malonylated flavonol glycosides, flavonol aglycones such as kaempferol and quercetin were never found in either previous study or the present work, implying their immediate conversion to downstream products and suggesting highly efficient in vivo synthesis of terminally modified flavonols [28,45]. Our findings indicate that the accumulation pattern of most flavonoids under drought showed a similar trend (elevated at 24 h while decreased at 48 h and increased again at 72 h) with the expression of key genes in the phenylpropanoids pathway, notably *MaPAL1*, *MaPAL2*, *MAPAL3*, *MaCHS1*, *MaF3H*, *MaF3GT*, and *MaUGT90A1*. A similar trend was also reported in *Achillea pachycephala* [22]. However, not all gene expression coincides with the accumulation trend of flavonoids, such as *MaC4H*, *MaF3′H1*, *MaF3G6*″*RT*, and *MaUGT76A2*. A possible explanation for this observation might be a homeostatic regulation of the phenylpropanoid pathway [46]. Furthermore, the metabolism of flavonoids is closely intertwined with metabolic signaling pathways that are activated in response to drought stress. A notable example is abscisic acid (ABA), a phytohormone associated with drought tolerance in plants. Gao et al. [9] have confirmed that the mutual regulation of ABA and flavonoid signaling contributed to the variation in drought resistance among different sea buckthorn subspecies. While the ABA content was not measured in this study, it is speculated that an increased ABA content would be evident based on findings from Liu’s research in mulberry [36].

The integration of multi-omics profiling and analysis has provided new insights into the biosynthesis studies of flavonoids and other bioactive compounds, such as 1-deoxynojirimycin (DNJ) in mulberry [13,47]. This study not only investigated the biochemical response of flavonoids under drought but also endeavored to identify potential key genes involved in flavonoid biosynthesis. The enzyme FLS catalyzes the conversion of dihydroflavonol to flavonol, serving as the initial key enzyme in the flavonol pathway. MaFLS1 protein characterized in this study catalyzed the conversion of dihydroflavonols to flavonols, potentially facilitating the enrichment of flavonol intermediators such as kaempferol and quercetin in mulberry. While a negative correlation was observed between the expression levels of *MaFLS2* and *MaFLS3* and flavonoid accumulation, the specific functions of these enzymes remained unexplored. Flavonols have various bioactivities beneficial to human health; however, their practical application is hindered by the high costs and extraction challenges. Compared to chemical synthesis and plant extraction, the idea of synthesizing flavonoids through fermentation using mulberry genes holds promise for producing secondary metabolites under controlled conditions [45]. This approach may potentially reduce production costs, avoid side effects, and mitigate environmental pollution problems [48].

Flavonoid-related UGTs in plants usually show strong substrate specificity [49]. In our study, the MaF3GT5 protein catalyzed the conversion of flavonol aglycones (e.g., kaempferol and quercetin) to flavonol 3-*O*-glucoside and flavonol 7-*O*-glucoside, but it did not utilize 3-*O*- and 7-*O*-glucosylated flavonols or UDP-rhamnose as potential substrates. Similarly, FaGT7 protein from strawberry (*Fragaria* × *ananassa*) primarily catalyzed the glycosylation at the 3-OH or 4′-OH position of kaempferol and quercetin, but it did not directly mediate the synthesis of flavonol di-*O*-glucoside [50]. A novel anthocyanidin 3-*O*-glucoside-2″-*O*-glucosyltransferase (In3GGT) was also reported in Japanese morning glory (*Ipomoea nil*), which recognized anthocyanidin 3-*O*-glucoside as substrates but not anthocyanidin 3, 5-di-*O*-glucoside [51]. Moreover, a genetic study of *AtUGT89C1* and *AtUGT79B6* in *Arabidopsis* observed that flavonol 7-*O*-rhamnosylation occurs after 3-*O*-glycosylation and glycosylation at 5-0H or 7-OH may occur after full modification at 3-OH [52,53], suggesting that several, as-yet uncharacterized *MaUGTs* involved in flavonol 3, 7-di-*O*-glucoside biosynthesis are encoded in the mulberry genome. Unlike the characterization of MaFLS1, a fermentation test was not performed in the functional study of the MaF3GT5 protein due to its lower efficiency of flavonoid production (Figure 6). To our knowledge, MaF3GT5 is the first UDP-glucosyltransferase identified in mulberry that forms 3-*O*- and 7-*O*-monoglucoside but not di-*O*-glucosides from flavonols such as kaempferol. Moreover, flavonoid biosynthetic and modification enzymes, such as flavonoid 3-*O*-glucoside-7-*O*-glucosyltransferase (3GGT) in mulberry, remain to be elucidated in further research.

## 4. Material and Methods

### 4.1. Plant Materials and Drought Treatment

The seeds of *Morus alba* var. *pendula* (a variety that belongs to *Morus alba* L. and can be stably cultivated in the lab) were stratified in bottles with water at 4 °C for 48 h to break dormancy. After vernalization, the seeds were disinfected with 0.1% HgCl_2_ for 10 min, followed by three washes with sterilized distilled water. Subsequently, the treated seeds were cultured in a growth chamber (watering with a modified Hoagland solution) under constant conditions (16 h light/8 h dark photoperiod, 70% relative humidity, and a culture temperature of 25 °C). When the seedlings reached 45 days, they were supplemented with 30% PEG-6000 to simulate drought stress. Leaf samples were then collected at 0, 24, 48, and 72 h of continuous treatment and immediately stored at −80 °C. Throughout the sampling period, regular water supply (once every 24 h) was maintained to ensure a consistent concentration of PEG-6000. To assess the impact of drought stress on the mulberry plants, the concentration of malondialdehyde (MDA), the activities of superoxide dismutase (SOD) and catalase (CAT) were analyzed after treatment. The content of SOD, MDA, and CAT were measured using the Grace reagent kit (Suzhou Grace Biotechnology Co., Ltd., Suzhou, China).

### 4.2. Sample Extraction for Metabolomic Analysis

The frozen mulberry leaves were pulverized into powder using a mixer mill (MM400, Retsch, Haan, Germany) with a zirconia bead for 1.5 min at 30 Hz. To extract flavonoids, 100 mg of the powder was weighed and then subjected to extraction with 0.5 mL 70% (*v*/*v*) aqueous methanol for 8 h (or overnight) at 4 °C. After centrifuged at 11,000× *g* for 10 min, 0.4 mL of the supernatant was filtered through a 0.22 μm filter (SCAA-104, ANPEL, Shanghai, China) and kept in 1.5 mL chromatographic sample bottles before LC-MS/MS analysis.

### 4.3. UHPLC-MS/MS Analysis Conditions

The sample extracts were analyzed using an LC-ESI-MS/MS system (UHPLC, Thermo Scientific™ Dionex™ UltiMate™ 3000 (Waltham, MA, USA); MS, Q Exactive hybrid quadrupole-orbitrap mass spectrometer; Thermo Fisher Scientific, Waltham, MA, USA). The UHPLC conditions were as follows: column, Aquity UPLC-BEH-C18 (1.7 μm particle size, length 2.1 × 150 mm^2^); solvent system, mobile phase A: ultrapure water (0.04% acetic acid), mobile phase B: acetonitrile (0.04% acetic acid); gradient program, 95:5 V_A_/V_B_ at 0 min, 5:95 V_A_/V_B_ at 20.0 min, 5:95 V_A_/V_B_ at 22.0 min, 95:5 V_A_/V_B_ at 22.1 min, 95:5 V_A_/V_B_ at 26.0 min; flow rate, 0.25 mL/min; column temperature, 40 °C; injection volume, 5 μL. The analytical MS/MS conditions were as follows: ESI source operation parameters: sheath gas, 35 arbitrary units; auxiliary gas, 10 arbitrary units; sweep gas, 0 arbitrary units; spray voltage, 3.5 KV; capillary temperature, 350 °C; and S-lens RF level, 50. The full MS parameters were as follows: MS scan range, 100–1000 *m/z*; resolution, 70,000; microscans, 1; automatic gain control (AGC) target, 1e6; Max IT, 200 ms. The data-dependent MS2 (dd-MS2) quantification method parameters were as follows: resolution, 35,000; microscans, 1; AGC target, 2e4; Max IT, 100 ms; loop count, 5; topN, 5; isolation window, 1.0 *m/z*; (N)CE: 15, 30, 60; apex-trigger, 2–6 s. Instrument tuning and mass calibration were performed with PierceTM LTQ Velos ESI positive ion calibration solution (Pierce, Rockford, IL, USA).

### 4.4. Transcriptome Sequencing, Assembly, and Analysis

Libraries were constructed from mulberry leaf mRNA and subjected to sequencing using the Illumina HiSeq™ 2500 platform by Gene Denovo Biotechnology Co. (Guangzhou, China). Transcriptome denovo assembly was carried out with a short reads assembling program—Trinity [54]. High-quality clean reads were obtained after removing sequences containing adapters, sequences with all A bases, sequences with more than 10% of unknown nucleotides (N), and sequences with over 50% of low-quality bases (Q-value ≤ 20). The reads, devoid of RNA (rRNA), from each sample were performed using a reference-free transcriptome analysis using TopHat2. Transcript reconstruction was performed using the Cufflinks 2.0.0 (Berkeley, CA, USA). Gene expression levels were normalized using the FPKM (fragments per kilobase of transcript per million mapped reads) method. Significantly differentially expressed genes (DEGs) were selected based on a fold change ≥ 2 and a false discovery rate (FDR) < 0.05. The DEGs were further subjected to GO functions and KEGG pathways enrichment analyses, with thresholds of *p* ≤ 0.01 and FDR ≤ 0.05 for both analyses. All unigenes were annotated by conducting BLASTx searches against the NCBI non-redundant protein (Nr) database, the Swiss-Prot protein database, the Kyoto Encyclopedia of Genes and Genomes (KEGG) database, and the COG/KOG databases, utilizing an E-value of 1 × 10^−5^. Protein functional annotations were obtained according to the best alignment results.

The in silico strategy to identify candidate flavonoid-related genes through a filtering procedure is as follows: first, transcripts were selected that are highly expressed in 24 h treated samples (D) versus the control (C) group. Subsequently, of the candidates identified, those possessing orthologs that are highly expressed in the treated group versus the control were selected. Finally, of the selected gene set, candidate flavonoid-related genes were selected according to the annotations and phylogenetic analysis.

### 4.5. Quantitative Real-Time PCR

Quantitative real-time PCR (qRT-PCR) was conducted using the Step One real-time PCR system (Applied Biosystems, Foster City, CA, USA). The gene-specific primers of *MaActin* and 18 phenylpropanoid biosynthetic genes were designed by Primer Premier 5 software and listed in Appendix A. Each qRT-PCR reaction was performed in a 20 μL mixture consisting of 1 μL of cDNA as template, 10 μL 2 × QuantiNova SYBR Green PCR Master Mix, 2 μL Rox dye (QiagenGmBH, Hilden, Germany), 0.6 μL of each primer, and DEPC-treated water. The amplification program consisted of an initial denaturation at 95 °C for 15 min, followed by 41 cycles of denaturation at 95 °C for 20 s, annealing at 55 °C for 40 s, and extension at 72 °C for 30 s. The 2^−ΔΔCt^ method was used to calculate the relative expression levels of the genes. All reported data are presented as means ± SD (*n* = 3).

### 4.6. Production of Recombinant Protein, In Vitro Enzyme Assay, and Fermentation Test

The full-length cDNA of *MaFLS1* and three *MaUGTs* were cloned into the pcold-TF expression vector and expressed as recombinant proteins in *Escherichia coli* strain BL21 (DE3). The bacterial culture was incubated at 37 °C until reaching an optical density at 600 nm of 0.6, followed by induction with a final concentration of 0.1 mM isopropyl-β-D-thiogalactopyranoside (IPTG) and further incubation at 25 °C for 8 h.

For the in vitro enzyme assays, cells were collected by centrifugation, and the fusion protein was purified using high-affinity Ni-NTA resin (L00250, GenScript, Nanjing, China) according to the manufacturer’s instructions. In the MaFLS1 reaction, 30 µL of purified MaFLS1 protein was dissolved in 100 mM phosphate buffer (pH 6.8) containing 100 µM dihydroflavonols substrate, 10 mM ascorbic acid, 10 mM 2-oxoglutarate, and 0.25 mM ferrous sulfate (FeSO_4_). In the MaUGTs reaction, 30 µL purified MaUGTs protein was dissolved in phosphate buffer saline (pH = 7.3) containing 100 µM flavonoid substrate, 100 µM UDP-glucose, and 100 µM UDP-rhamnose. Both reactions were incubated at 28 °C for 30 min with continuous agitation. The fermentation process mirrored that previously utilized for characterizing *MaCHS* [45]. Cells were harvested by centrifugation, resuspended in M9 medium containing 0.1 mM IPTG, 0.1 mg/mL ampicillin, and 1 mM dihydroflavonols, and incubated with shaking at 25 °C for 48 h. Flavonoids in the resulting solution and the final fermentation product were extracted with 70% (*v*/*v*) aqueous methanol, filtered through a 0.22 μm filter, and subjected to LC-MS/MS analysis.

### 4.7. Chemicals and Standards

All solvents and reagents used in this study were of analytical grade. Methanol, acetonitrile, and acetic acid (HPLC grade) were purchased from Thermo-Fisher Scientific (Shanghai, China). Authentic standards, including kaempferol, quercetin, 3-*O*- and 7-*O*-glycosylated flavonols, kaempferol 3-*O*-rutinoside (K3R), and rutin were purchased from Chroma Biotechnology Co., Ltd. (Chengdu, China). Dihydrokaempferol and dihydroquercetin were purchased from DeSiTe Biological Technology Co., Ltd. (Chengdu, China). Other standards were purchased from Sigma-Aldrich (Shanghai, China). UDP-glucose and UDP-rhamnose were purchased from EnzymeWorks Inc. (Suzhou, China).

### 4.8. Statistical Analyses

The metabolites and RNA-seq data were analyzed using R software (version 4.1.3), incorporating heat-map analysis, principal component analysis (PCA), and multiple testing. Figures were generated with SigmaPlot 12.0 (Systat, San Jose, CA, USA) and Adobe Illustrator CS6 (Adobe, San Jose, CA, USA). The results are presented as means ± standard deviation (SD). The calculated *p*-value was calibrated using false discovery rate (FDR) correction.

## 5. Conclusions

In this study, we conducted a comprehensive investigation of the biochemical response of flavonoids in mulberry leaves under drought conditions through multi-omics profiling and analysis. Our findings have enhanced our understanding of the phenylpropanoid pathway in mulberry. Specifically, we observed that drought stress led to a significant increase in the activity of protective enzymes upregulated the expression of genes involved in the phenylpropanoid pathway, and stimulated the biosynthesis of flavonoids. Additionally, we identified and characterized one *MaFLS* and three flavonoid-related UPD-glycosyltransferases by analyzing transcriptome data. The recombinant MaFLS1 and MaF3GT5 proteins were found to catalyze the synthesis of flavonol aglycones and flavonol *O*-monoglucoside, respectively. These results represent a significant advancement in our knowledge of flavonol biosynthesis in mulberry and provide valuable insights into the biosynthesis of other bioactive compounds.

## Figures and Tables

**Figure 1 ijms-25-07417-f001:**
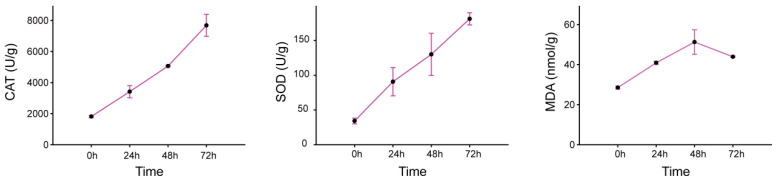
Variation in catalase (CAT), superoxide dismutase (SOD), and malondialdehyde (MDA) levels at different hours under drought stress in *Morus alba* var. *pendula* Dippel leaves. Data represent the mean of three replicates with standard deviation (±SD).

**Figure 2 ijms-25-07417-f002:**
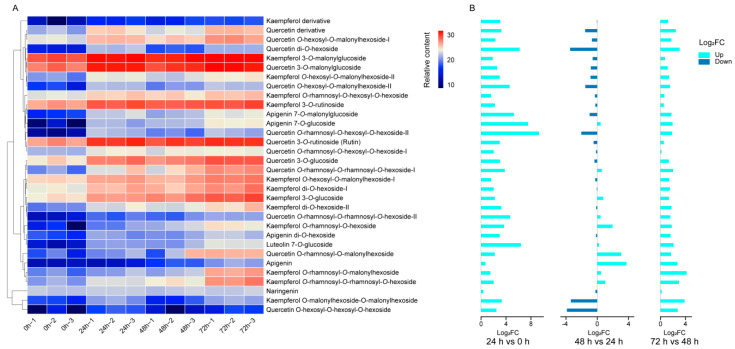
Metabolic profiling and flavonoid variation in *M. alba* var. *pendula* leaves under drought stress. (**A**) LC/MS profiling of flavonoids at 0, 24, 48, and 72 h in mulberry leaves under drought treatment, each with three biological replicates. The heat-map shows values displayed on log2 of the relative peak area. (**B**) Fold changes of flavonoids at 24, 48, and 72 h. Increased flavonoids are indicated in cyan color, while decreased flavonoids are indicated in blue.

**Figure 3 ijms-25-07417-f003:**
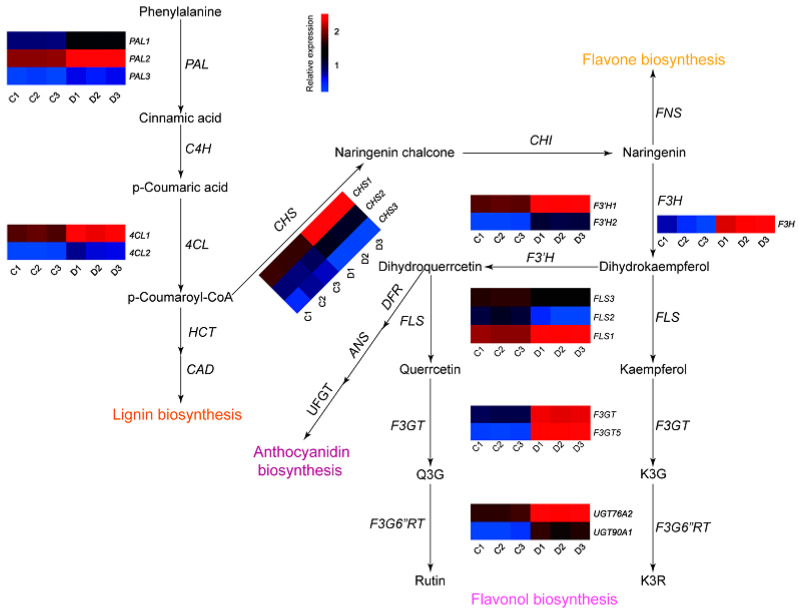
The proposed flavonoid pathway and heat-map of key genes involved in the drought response of mulberry leaves. Colored blocks indicated different relative expression levels (from blue to red). C1 to C3 represents three replicates of the control, while D1 to D3 represents the drought-treated group. Abbreviations: *PAL*, phenylalanine ammonia lyase; *C4H*, cinnamate 4-hydroxylase; *4CL*, 4-coumarate-CoA ligase; *CHS*, chalcone synthase; *CHI*, chalcone isomerase; *FNS*, flavone synthase; *F3H*, flavanone 3-hydroxylase; *F3*′*H*, flavanone 3′-hydroxylase; *FLS*, flavonol synthase; *F3GT*, flavonoid 3-O-glycosyltransferase; *F3G6”RT*, flavonol 3-*O*-glucoside: 6″-*O*-rhamnosyltransferase; K3G, kaempferol 3-*O*-glucoside; Q3G, quercetin 3-*O*-glucoside; K3R, kaempferol 3-*O*-rutinoside.

**Figure 4 ijms-25-07417-f004:**
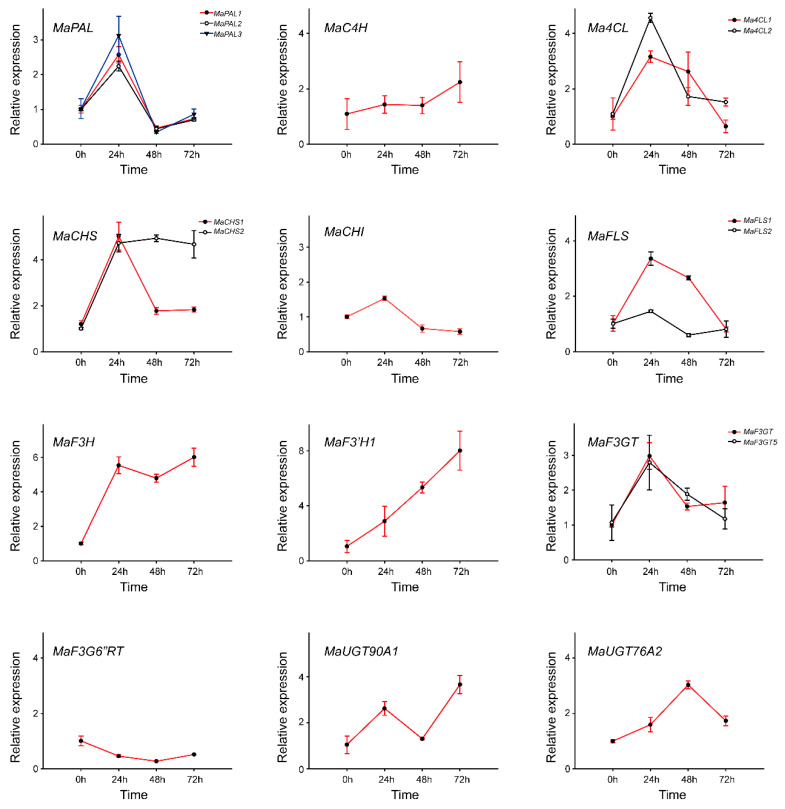
Transcript levels of phenylpropanoid biosynthetic genes and candidate genes under PEG-6000-induced drought stress. Data represent the mean of three replicates with standard deviation (±SD).

**Figure 5 ijms-25-07417-f005:**
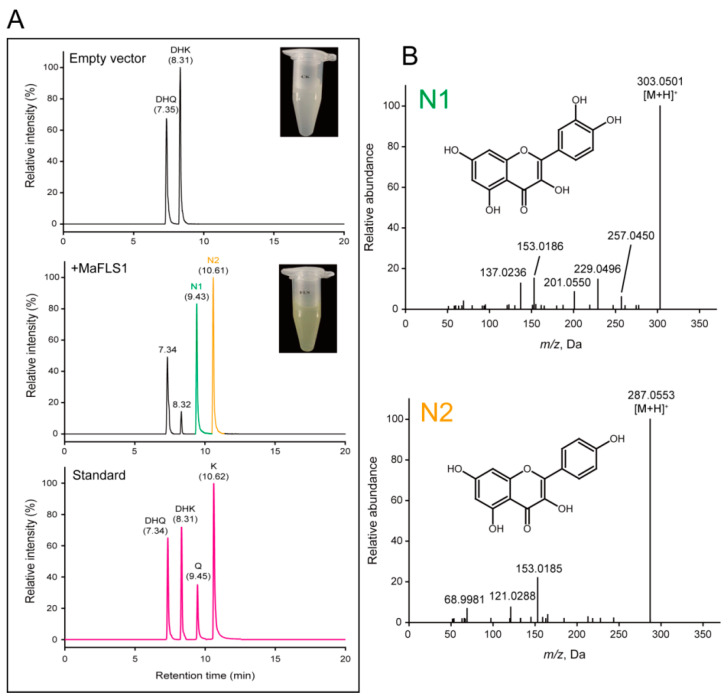
UHPLC-MS/MS analyses of the reaction of recombinant MaFLS1 protein and fermented products. (**A**) Elution profile of the reaction products of His tag protein (empty vector), MaFLS1 protein (+MaFLS1), and authentic standards. The chromatograms of fermentation extracts present a similar pattern to the in vitro enzyme assay. (**B**) Extracted fragment mass chromatograms of reaction products (N1 and N2). Abbreviations: DHK, dihydrokaempferol; DHQ, dihydroquercetin; K, kaempferol; Q, quercetin.

**Figure 6 ijms-25-07417-f006:**
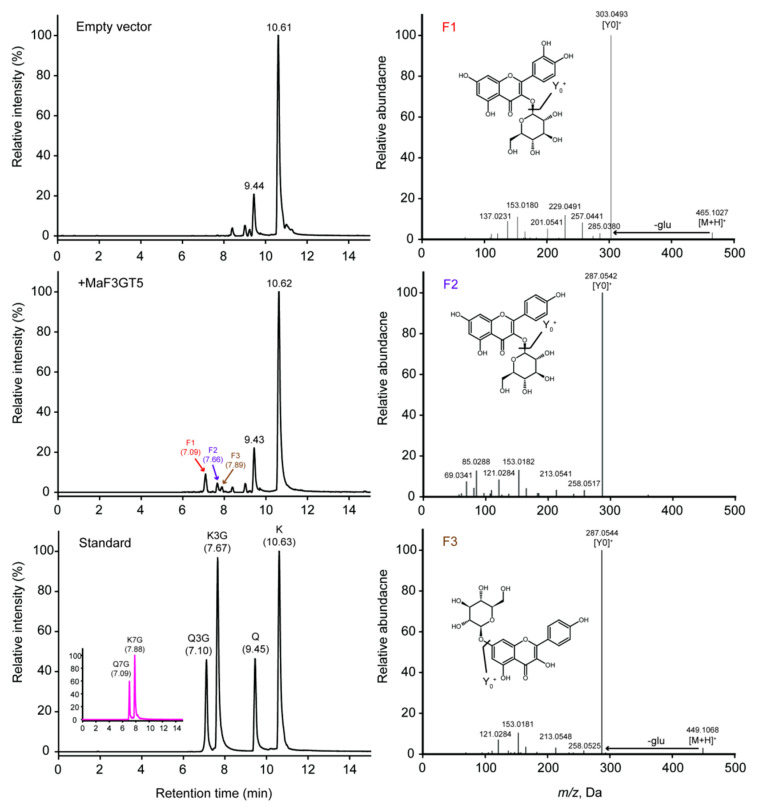
HPLC-MS/MS analyses of the reaction of recombinant MaF3GT5 protein. The elution profile of the reaction products involving His tag protein (empty vector), MaF3GT5 protein (+MaF3GT5), and chromatograms of standards (K, Q, K3G, Q3G, K7G, and Q7G) are shown on the left. Extracted fragment mass chromatograms of reaction products (F1, F2, and F3) are shown on the right. Abbreviations: K, kaempferol; Q, quercetin; K3G, kaempferol 3-*O*-glucoside; Q3G, quercetin 3-*O*-glucoside; K7G, kaempferol 7-*O*-glucoside; Q7G, quercetin 7-*O*-glucoside; -glu, natural loss of glucoside.

## Data Availability

The data presented in this study are available on request from the corresponding author.

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
