# Peer review of "Metabolic and Transcriptional Analysis Reveals Flavonoid Involvement in the Drought Stress Response of Mulberry Leaves"

_ijms, 2024, doi:10.3390/ijms25137417_

Round 1
Reviewer 1 Report
Comments and Suggestions for Authors
This paper reports a comprehensive study of the biochemical response of flavonoids in mulberry leaves under drought conditions. Drought stress was imposed on 45days-old mulberry seedlings by 30% PEG 6000 in the nutrient solution for 72 hours (short term stress, rather severe). The level of the stress was followed by measuring the activities of the antioxidant enzymes catalase and superoxide dismutase and MDA content in leaves. Metabolic profiling of flavonoids by LC/MS was made in dynamics at three time points, and the expression of key genes related to flavonoid biosynthesis was analysed through RNA sequencing. Selected genes of the phenylpropanoid pathway were further analysed through qRT-PCR. Recombinant putative Morus alba L. flavonol synthase1 and flavonoid-related UGTs were purified and the products of their enzymatic reactions were identified by UHPLC-MS/MS analysis as quercetin and kaempferol and their glycosides.
The paper is written clearly and concisely with all necessary details and in a proper English. I have only minor remarks.
Line 149 - dertain flavonoids or certain flavonoids?
Line 216 – repetition of “genes”
Lines 310-311 “The MDA content serves as an indicator of stress severity. Although the MDA accumulation presented a certain degree of decrease at 72 hours… “ – The decrease of MDA may be due to the efficient work of ROS detoxifying enzymes, especially CAT, which activity constantly increases during the tested time period. The increase in the content of secondary metabolites may also contribute for taming the oxidative damage to membranes, an indicator of which is MDA.
Line 339-341 “despite naringenin being considered an intermediate and rapidly catalyzed as the common precursor of the large number of downstream flavones and flavonols, its content remained at a relatively low and almost unchanged level” – as naringenin is a common precursor and is rapidly converted, it is logical that its level would not change significantly, it undergoes quick conversion into other secondary metabolites
Line 415 – concentration of SOD and CAT – actually activity was measured
Figure S1 – Please check the ABCD letters – they should correspond to the description, but the figure begins with a Venn diagram
Table S1 and fig 4 – The genes correspond to the text in the results but in fig 4 the transcript changes of 12 genes are presented and the primers for qRT-PCR are 18 – what was the reason for this?
Reviewer 2 Report
Comments and Suggestions for Authors
This study chose two genes in terms of their functional analysis after analyzing transcriptomic and metabolomic responses to drought stress. The authors need to perform more comprehensive analysis for those two genes to show their orthologies as they are members of a large gene family. Also, method section should be improved to describe the procedures. Please refer to the comments included in the attached file.

Comments on the Quality of English LanguageThere are some grammatical errors in the manuscript. Please revise the manuscript carefully.
